# Development of a Fluorescent Tool for Studying *Legionella bozemanae* Intracellular Infection

**DOI:** 10.3390/microorganisms9020379

**Published:** 2021-02-13

**Authors:** Breanne M. Head, Christopher I. Graham, Teassa MacMartin, Yoav Keynan, Ann Karen C. Brassinga

**Affiliations:** 1Department of Medical Microbiology and Infectious Diseases, University of Manitoba, Winnipeg, MB R3T 2N2, Canada; Umheadb@myumanitoba.ca; 2Department of Microbiology, University of Manitoba, Winnipeg, MB R3T 2N2, Canada; Graham13@myumanitoba.ca (C.I.G.); Macmartt@myumanitoba.ca (T.M.); Ann.brassinga@umanitoba.ca (A.K.C.B.)

**Keywords:** *Acanthamoeba castellanii*, GFP, intracellular infection, *Legionella*, *Legionella bozemanae*, non-pneumophila

## Abstract

Legionnaires’ disease incidence is on the rise, with the majority of cases attributed to the intracellular pathogen, *Legionella pneumophila.* Nominally a parasite of protozoa, *L. pneumophila* can also infect alveolar macrophages when bacteria-laden aerosols enter the lungs of immunocompromised individuals. *L. pneumophila* pathogenesis has been well characterized; however, little is known about the >25 different *Legionella* spp. that can cause disease in humans. Here, we report for the first time a study demonstrating the intracellular infection of an *L. bozemanae* clinical isolate using approaches previously established for *L. pneumophila* investigations. Specifically, we report on the modification and use of a green fluorescent protein (GFP)-expressing plasmid as a tool to monitor the *L. bozemanae* presence in the *Acanthamoeba castellanii* protozoan infection model. As comparative controls, *L. pneumophila* strains were also transformed with the GFP-expressing plasmid. In vitro and in vivo growth kinetics of the *Legionella* parental and GFP-expressing strains were conducted followed by confocal microscopy. Results suggest that the metabolic burden imposed by GFP expression did not impact cell viability, as growth kinetics were similar between the GFP-expressing *Legionella* spp. and their parental strains. This study demonstrates that the use of a GFP-expressing plasmid can serve as a viable approach for investigating *Legionella* non-pneumophila spp. in real time.

## 1. Introduction

*Legionella* are Gram-negative, rod-shaped bacteria that cause Legionnaires’ disease (LD), which is a severe type of pneumonia that remains a global and public health concern. In recent years, cases of Legionnaires’ disease have increased at an alarming rate, with outbreaks having been reported across North America, Europe, and Australia [1,2,3]. Although the exact reason for this increase is unknown, it is likely multifactorial, representing increased disease awareness and diagnostic testing, in addition to an increase in the susceptibility of the global population, as well as global warming, and aging infrastructure, both of which favor *Legionella* growth in the environment. In Canada, *Legionella* incidence has risen 10-fold since 2000, with 635 cases having been reported to the Public Health Agency of Canada in 2018 [4]. Incidence has also increased 9-fold in the United States (US) and 2-fold in Europe [5,6]. *Legionella* cases are thought to be under-reported for reasons discussed later. A study by Collier et al. [7] estimated that *Legionella* species (spp.) caused 10,800 (95% confidence interval (CI) 7280–13,100) patient hospitalizations in the US in 2014, equating to $401 million (95% CI $79–1690) in total direct healthcare costs, highlighting the significant burden that these organisms have on the healthcare system. 

In the environment, *Legionella* spp. can be found in naturally occurring and man-made water systems such as showers, cooling towers, fountains, lakes, and streams [8]. To survive these low-nutrient habitats, they have adapted to parasitize a variety of protozoan species including *Vermamoeba*, *Naegleria,* and *Acanthamoeba*. Once inside their protozoan host, *Legionella* can subvert the host cellular machinery by creating a replicative niche, known as a *Legionella*-containing vacuole (LCV), where they can proliferate to high titers until the cell ruptures, facilitating their release back into the environment [9,10,11].

The *Legionella* genus is comprised of more than 50 different species, with half of them able to cause disease in humans [12]. However, most available information about *Legionella* spp. pathogenesis and interaction with its host stems from studies focused on *Legionella pneumophila* serogroup (SG) 1, which is the type strain [13]. Diagnostics, although previously reliant on culture, have shifted toward the use of the urinary antigen test (UAT). The UAT is a rapid and inexpensive method that has high sensitivity for *L. pneumophila* SG1, favoring the detection of this organism while leading to false-negative results when patients are infected with other *L. pneumophila* SG or *Legionella* spp. [14]. Indeed, studies from North America and Europe, regions which tend to favor the use of the UAT for diagnosing *Legionella* spp. infections, report *L. pneumophila* SG1 as the primary cause of LD (>90% of cases) [15]. This may be more reflective of the diagnostic methodologies rather than a true representation of species diversity. In a study conducted by Ng et al. [16], which analyzed 1401 cases of *Legionella* infection between 1978 and 2006, it was found that when culture was used, only 66% of cases were, in fact, *L. pneumophila* SG1. The authors suggested that proximity to the Great Lakes played a role in disease epidemiology, which further supports the findings that *Legionella* epidemiology can vary by region. Indeed, in places such as Australia and New Zealand, other *Legionella* spp. such as *L. longbeacheae* are more prevalent in the environment and have been shown to account for up to half of all of the *Legionella* cases in Australia [2,15]. Consequently, alternative methods such as serology or polymerase chain reaction (PCR) are more commonly used for diagnosing *Legionella* infections in some of these regions [17]. 

Infections due to *Legionella* non-pneumophila spp., such as those caused by *L. bozemanae*, have most frequently been reported among individuals that have underlying immune suppression, which is associated with malignancy or HIV infection [18], as well as recipients of immunomodulatory treatments [19,20]. Aside from causing pneumonia, *L. bozemanae* infection can lead to cavitary lung lesions [21], soft-tissue infections [22], and septic arthritis [21]. However, diagnosing *L. bozemanae* infections is difficult, since the UAT and serological kits currently do not target *L. bozemanae*, and traditional culture techniques often result in non-detection due to the use of cefamandole-containing media, which can inhibit the growth of non-pneumophila spp. [21,23]. Consequently, infections due to *L. bozemanae* have primarily been reported as case studies, and information on *L. bozemanae* virulence and pathogenesis is currently lacking.

Since infection models are well established for *L. pneumophila* SG1 pathogenesis, we sought to use these models to glean insight into *Legionella* non-pneumophila infections. As fluorescent-based detection systems have been used extensively to investigate host–cell interactions, bacterial pathogenesis, and protein function in *L. pneumophila* studies and in other bacterial species [24,25,26,27], we aimed to exploit their use for studying *L. bozemanae*. The green fluorescent protein (GFP) reporter system is a well-documented tool that can be used for fluorescence, since it is bright enough to detect above autofluorescence levels and stable enough to allow measurements over time [28,29]. However, it is important to note that optimal use of a GFP-expressing plasmid is best achieved if the strain of interest is verified to be devoid of other plasmids. *Legionella* species have been reported to have naturally occurring plasmids that may be incompatible with the IncQ-group derived GFP plasmid once introduced [30]. Moreover, although different bacterial species appear to tolerate GFP better than others, the overexpression of GFP in some bacteria can cause toxicity [31]. 

Here, we describe the development of a GFP reporter plasmid that was transformed into two *Legionella* clinical isolates *(L. bozemanae* and *L. pneumophila*) and the type strain, *L. pneumophila* American Type Culture Collection (ATCC) 33152. This study serves as a tool to create fluorescent *Legionella* strains, which can be used to characterize non-pneumophila *Legionella* bacterial growth kinetics in vitro (broth) and in vivo (*Acanthamoeba castellanii* infection model).

## 2. Materials and Methods 

### 2.1. Bacteria, Plasmids, Primers, and Growth Conditions

Bacterial strains and plasmids used in this study are listed in Table 1. Primers and amplicon product sizes are listed in Table 2. *Escherichia coli* DH5α strains were grown in Luria–Bertani broth or on agar, supplemented with kanamycin (40 μg/mL) or ampicillin (100 μg/mL) where appropriate, and incubated overnight at 37 °C. *Legionella* spp. were streaked from frozen stocks (Buffered Yeast Extract (BYE) + 10% dimethyl sulfoxide) onto Buffered Charcoal Yeast Extract (BCYE) agar at 37 °C with 5% CO_2_ for 3–4 days then if required, heavily inoculated into BYE broth [32] for 18–24 h at 37 °C on a tube rotator, supplemented with kanamycin (25 μg/mL). 

### 2.2. Construction of GFP-Expressing Legionella Strains

In general, *L. pneumophila* can maintain RSF1010-based plasmids [34,35]. In this view, pKB127 was genetically modified to replace the *tdΔI* marker with a kanamycin resistance marker, sourced from p34S, creating pGFP-Kan for use with the *Legionella* strains in this study. pKB127 refers to the insertion of a 245 base pair (bp) *magA* promoter region in the promoterless *gfp* reporter plasmid pBH6119 [26] and was chosen, since *magA* is a strong promoter that tends to be more expressed post-exponentially. pBH6119 carries a thymidylate synthase *(tdΔI*) that rescues the thymidine auxotrophy of Lp02, which is a derivative of *L. pneumophila* Philadelphia-1 [36], thereby providing a convenient non-antibiotic-based plasmid selection. However, since *Legionella* spp. are generally not thymidine auxotrophs, a kanamycin resistance marker was used as a substitute. 

Clinical isolates of *L. bozemanae* and *L. pneumophila*, along with *L. pneumophila* ATCC 33,152 type strain, were confirmed to lack intrinsic resistance to kanamycin by plating on BCYE supplemented with kanamycin and checked for growth daily for 96 h. 

To generate GFP-expressing *Legionella* strains, the P_magA_*gfp* reporter plasmid (pKB127) was genetically modified by replacing the *tdΔI* selection marker with a kanamycin resistance marker. Briefly, pKB127 was cut with EcoRI and SacII to drop out the *tdΔI* selection marker, with the PCR-amplified kanamycin resistance marker ligated in its place, creating pGFP-Kan. *E. coli* DH5α was used as the host strain for cloning approaches. Amplification of the kanamycin resistance marker with engineered restriction sites was achieved with primer set PF KanR EcoRI/PR KanR SacII and template p34S using Q5 High Fidelity Polymerase (New England Biolabs, Canada). The construct of pGFP-Kan was verified by polymerase chain reaction (PCR) using a combination of the internal and external primer sets listed in Table 2. Then, the plasmid was electroporated into the *L. bozemanae*, *L. pneumophila* clinical, and *L. pneumophila* ATCC 33,152 type strains according to the methods described in Tanner et al. [35] to create the strains LP 1031, LB 1032, and LP 1030, respectively, which were then plated on BCYE supplemented with kanamycin to select for the kanamycin resistance phenotype. Plasmid integration was confirmed by screening the kanamycin-resistant GFP-expressing *Legionella* strains with the PCR confirmation primers (Table 2). 

### 2.3. In Vitro Growth Kinetics and Relative Fluorescence Assays

In vitro growth of the *Legionella* parental and GFP-expressing strains was assessed in BYE broth as described previously [32,37]. Briefly, each strain was grown on the appropriate BCYE medium for 96 h and then resuspended in BYE, either with or without kanamycin, and inoculated into a 96-well plate at a final concentration of 1.5 × 10^7^ cells/mL [35]. The plate was incubated for 24 h at 37 °C with shaking in a BioTek Synergy HTX multi-mode automated microplate reader and optical density (600 nm; OD_600_) and GFP relative fluorescence (excitation and emission wavelengths 485/20 nm and 528/20 nm, respectively) was measured every hour. Media background fluorescence and absorbance was subtracted from the relative fluorescence unit (RFU) values and optical density (OD_600_) data, respectively, and values for each of the GFP strains along with their parental strains were plotted over time. 

### 2.4. Intracellular Growth Assays in Acanthamoeba castellanii

For maintenance of *A. castellanii* ATCC 30234, amoeba were cultured in ATCC 712 medium, Peptone Yeast Extract Glucose (PYG) liquid medium, at 25 °C [26]. Amoeba were split every 48–96 h (depending on the initial seeding dilution). For infections, a 48-h culture of *A. castellanii* was washed and resuspended in *A. castellanii* buffer (AC buffer; 4 mM magnesium sulfate, 0.4 mM calcium chloride, 3.4 mM sodium citrate dihydrate, 0.05 mM ferrous ammonium sulfate hexahydrate, 2.5 mM disodium phosphate, and 2.5 mM potassium dihydrogen phosphate) and then seeded into 24-well plates at a density of 1 × 10^6^ cells/mL [10]. Once cells were adherent (≈2 h at 25 °C), bacteria (from 3–4-day-old BCYE-agar plates) were suspended in AC buffer, and titers were determined at an OD_600_. Then, amoeba were infected with *Legionella* parental and GFP-expressing strains at a multiplicity of infection (MOI) of 0.1. Following a 1 h incubation at 25 °C, to synchronize the infection, amoeba were washed 3 times in AC Buffer to remove any extracellular bacteria that had not been taken up into the cells [35]. At various time points (1, 48, 144, and 216 h post-infection), amoeba were lysed using mechanical disruption (i.e., by scraping the monolayer, vortexing the sample for 30 s followed by centrifugation for 8 min at 14,800 rpm) [38], and bacterial counts were determined by serial dilution BCYE plate counts (colony-forming units (CFU)/mL). 

### 2.5. In Vitro and In Vivo Microscopic Imaging of the GFP-Expressing Legionella Strains

All GFP-expressing *Legionella* strains were cultured on BYCE agar supplemented with kanamycin for 96 h prior to in vitro and in vivo imaging studies. For in vitro imaging, a single isolated colony was resuspended in 100 μL of BYE and mounted on a glass microscope slide. Then, bacteria were heat-fixed (i.e., by placing a drop of bacteria on a slide, letting it dry, and then passing the slide through a flame 2 times) and mounted in ImmunoMount medium (Thermo Scientific, Waltham, MA, USA) before viewing.

To visualize GFP-expressing *Legionella* within *A. castellanii*, the day prior to infection, amoeba were placed in a 6-well plate containing sterile poly-lysine coated coverslips (Neuvitro Corp, Vancouver, WA, USA) at a density of 3 × 10^6^ cells/well and allowed to adhere overnight in a 25 °C incubator. Then, *A. castellanii* was infected with each GFP-expressing strain at an MOI of 5, and the infection was incubated for 6 or 12 h. Following infection, the cells were washed 3 times with AC buffer, stained with an ER-Tracker Red dye (BODIPY TR Glibenclamide; ThermoFisher Scientific, Whitby, ON, Canada) for 30 min at 37 °C, and fixed in formaldehyde. Then, the coverslips containing the infected amoeba were mounted onto glass slides with ProLong Gold Antifade Mountant with DAPI (ThermoFisher Scientific, Canada) per the manufacturer’s instructions. 

For fluorescence imaging, all slides were examined by differential interference contrast (DIC) and confocal microscopy using an LSM 700 confocal microscope with accompanying Zeiss software. GFP-expressing *Legionella*, the ER-Tracker Red Dye, and DAPI had an excitation and emission spectra of 488/583, 587/615, and 405/461 nm, respectively. Images were generated using a 40× oil immersion objective. All fluorescent images were edited using ImageJ version 1.52A [39].

### 2.6. Statistical Analyses

All experiments were performed in triplicate unless otherwise stated. Data were analyzed using STATA® version 14. Bacterial optical densities and growth (CFU/mL) are reported as the mean and standard error. To determine if optical densities were similar between the fluorescent strains and their parental counterparts at inoculation, a T-test was performed, while a one-way analysis of variance was used to compare optical densities between the *Legionella* parental strains (*L. bozemanae* (948), *L. pneumophila* clinical (950) and *L. pneumophila* ATCC 33152) as well as those from the GFP-expressing strains (*L. bozemanae* (LB 1032), *L. pneumophila* clinical (LP 1031) and *L. pneumophila* ATCC 33,152 (LP 1030)). To compare bacterial optical densities and fluorescence intensity over time between the GFP-*Legionella* strains, a repeated-measures analysis of variance was carried out with a Bonferroni post-hoc analysis conducted when data were significant (*p* ≤ 0.05). All data were plotted using GraphPad Prism version 9 (GraphPad Software, San Diego, CA, USA).

## 3. Results

### 3.1. Generation of GFP-Expressing Legionella Strains

Transformation of the *Legionella* strains by electroporation was carried out with relative ease, requiring one attempt when constructing the GFP-*L. pneumophila* strains and two attempts for the GFP-*L. bozemanae* strain. Following electroporation and plating on BCYE supplemented with kanamycin, transformants were streaked out on fresh agar plates for purity, after which frozen stocks were prepared. Fluorescence microscopy confirmed GFP expression (Figure 1). Colony characteristics of the GFP isolates, such as colony shape and size, did not appear to differ from their parental strains. 

### 3.2. Plasmid-Borne GFP Expression Does Not Affect the Growth of Legionella In Vitro

To determine if the presence of pGFP-Kan affects *Legionella* growth profiles, bacterial kinetic studies were performed to compare the growth rates of each of the GFP-expressing *Legionella* strains with their parental counterparts. Each of the *Legionella* pairs (with and without pGFP-Kan) displayed similar growth curves over the 24 h period with all of the strains reaching their peak density between 12 and 16 h (Figure 2A–C). No significant differences were seen between the optical densities of the parental strains and those of their GFP-expressing counterparts.

When comparing the growth profiles of the *L. pneumophila* parental and transformant strains with that of the *L. bozemanae* strains, higher densities were seen among the *L. pneumophila* species compared to those seen with *L. bozemanae* (OD_600_ of 1.0 vs. 0.5, respectively, *p* < 0.0001). Likewise, the GFP-expressing *L. pneumophila* strains also displayed higher rates of fluorescence compared to those of the GFP-expressing *L. bozemanae* (*p* < 0.0001), with 24-h RFU values of 3385, 3137, and 857 for strains LP 1030, LP 1031, and LB 1032, respectively (Figure 2D–F). None of the *Legionella* parental strains were fluorescent as evidenced by the RFU values, which mimicked that of the BYE media control. 

Similarly, when RFU values were normalized to optical density, *L. bozemanae* fluorescence was consistently lower than that of the GFP-expressing *L. pneumophila* strains (Figure 2G). Indeed, the initial GFP activity of the LB 1032 strain began at approximately 2000 RFU/OD_600_ units, while activity was >3000 RFU/OD_600_ units for the LP 1031 and LP 1030 strains. Each of the GFP-expressing strains displayed a similar trend over the 24-h experiment: the RFU/OD_600_ values decreased throughout the first 7–10 h of the assay, at which time the RFU/OD_600_ values rebounded, increasing gradually until they returned to their approximate starting values. The growth-dependent GFP expression profiles exhibited by all three strains reflect the activity of the *magA* promoter, which tends to be higher post-exponentially as reported elsewhere [26].

### 3.3. Plasmid-Borne GFP Expression Does Not Affect Legionella Growth within A. castellanii protozoa

To determine if plasmid-borne GFP expression altered *Legionella* growth and viability in vivo, *A. castellanii* protozoa were infected with each of the *Legionella* strains, and intracellular growth kinetics were assessed. *Legionella* growth profiles appeared to be unaffected by GFP expression. Both parental and GFP-expressing *L. pneumophila* strains (ATCC 33,152 and clinical) showed similar growth profiles, achieving 10^7^ CFU/mL by the 216-h time point (Figure 3). Likewise, the GFP-expressing *L. bozemanae* strain also exhibited a similar growth profile in vivo compared to its parental counterpart.

However, as was seen with the in vitro experiments, *L. bozemanae* growth was less efficient than *L. pneumophila*, which was highlighted by the lower log titers seen with the *L. bozemanae* strains (*p* < 0.001). At each time point, *L. bozemanae* growth was approximately 1 log lower than the *L. pneumophila* species, with *L. bozemanae* achieving 10^5^ and 10^6^ CFU/mL by the end of the 216-h experiment for the parental and fluorescent strain, respectively. These results indicate that this non-pneumophila spp. may not be as adept as *L. pneumophila* for intracellular uptake or replication within protozoa. 

### 3.4. The Intracellular Life Cycle of L. bozemanae Is Similar to That of L. pneumophila

The intracellular lifestyle of *L. pneumophila* in protozoa is well established; subversion of the digestive pathway and formation of the replicative vacuole is dependent on the Dot/Icm pathway [40]. However, very little is known about the intracellular lifestyle of non-pneumophila species, such as *L. bozemanae*. 

To gain further insight into *Legionella* uptake, growth rates, and fluorescence over time, we visualized *Legionella* infection in *A. castellanii* using confocal microscopy. At 6 h post-infection, bacteria can be seen within the host (Appendix A); however, *A. castellanii* infected with the *L. pneumophila* spp. tended to have more than 1 bacterium present per cell. At 12 h post-infection, the bacteria can be seen clustered together and dividing inside the cell, which is a process that appeared to be similar for both species (Figure 4).

## 4. Discussion

To date, research regarding *Legionella* has rarely focused on species outside of *L. pneumophila* SG1. Here, we report, for the first time, the generation of a GFP-expressing *L. bozemanae* strain derived from a clinical isolate obtained from our local hospital. Transformation did not appear to affect the growth kinetic profiles of the GFP-expressing *Legionella* strains compared to their parental counterparts. 

This work extends our knowledge of non-pneumophila species by comparative studies of the growth profiles and relative fluorescence of *L. bozemanae* to *L. pneumophila*. Since the *Legionella* genus encompasses a diverse group of organisms, it is important to expand research to include non-pneumophila spp., particularly those that can cause disease [41,42,43]. In our study, several similarities and differences between *L. pneumophila* and *L. bozemanae* were seen. In broth, although each of the strains grew well in BYE at 37 °C, the *L. pneumophila* strains featured a faster growth rate with higher optical densities than that observed for *L. bozemanae*. These results match those observed in earlier studies [14,15,44], which reported that the non-pneumophila spp. often exhibit slower and reduced growth rates compared to *L. pneumophila* when grown in vitro at 37 °C. 

In the protozoan model, the GFP-expressing *L. bozemanae* and *L. pneumophila* clinical strains, as well as the fluorescent *L. pneumophila* ATCC 33,152 strain, were all able to infect and replicate within *A. castellanii*. As amoeba share many similarities to human phagocytic cells, it was not surprising that the *L. bozemanae* clinical isolate was able to infect and replicate in the amoeba model. Although the GFP-expressing *L. bozemanae* strain did not attain growth levels as high as the GFP-expressing *L. pneumophila* strains, a steady increase in bacterial load was seen throughout the experiment. These findings differ from those of Neumeister and co-workers [45], which reported that *L. bozemanae* was not able to multiply within *A. castellanii* and rather showed a decreasing bacterial count in amoeba following infection. A possible explanation for these differing results may be attributed to strain differences as variability regarding infection rates, replication efficiency, and pathology has been documented, even within a single species [42,45,46]. Moreover, the lack of standardization regarding experimental conditions for *Legionella* infections in the *A. castellanii* infection model may also be a contributing factor. 

Next, confocal microscopy was used to visualize *Legionella* spp. intracellular growth inside *A. castellanii* over time. *L. bozemanae* bacteria appear to replicate and cluster together, which may be indicative of vacuole formation, suggesting that the intracellular lifestyle of *L. bozemanae* may be similar to that of *L. pneumophila*. However, these findings need to be further defined through the use of detailed microscopy studies of *L. bozemanae* strains that are dysfunctional for genetic components known to be associated with pathogenesis as characterized by *L. pneumophila.*

Comparing the fluorescent intensities between the *Legionella* strains showed that the *L. pneumophila* strains, as comparative controls, featured stronger fluorescence than *L. bozemanae* even when normalizing fluorescence by optical density. Heterogeneity in fluorescence signals can occur due to several factors including differences in the copy numbers of the plasmids inside the cells, differences in plasmid protein expression, or poor GFP folding [47,48]. Nevertheless, the matched GFP expression profiles between the *Legionella* strains in this study with that previously reported by Hiltz and colleagues [49] indicate the recognition of a promoter region sourced from *L. pneumophila* by transcription and translation machinery in *L. bozemanae*, thereby allowing GFP expression. Despite the lower levels of GFP expression in *L. bozemanae*, the bacterial cells were bright enough to be seen clearly during intracellular infections of *A. castellanii*, and they can be used for further investigations of host–pathogen interactions. Further work is required to refine this approach by optimizing GFP expression through further genetic modifications of the plasmid as well as conducting whole genome sequencing to check for presence of any naturally occurring plasmids.

Taken together, this study shows the feasibility of using plasmid-borne GFP expression as a tool to investigate the intracellular lifestyle of lesser-known non-pneumophila *Legionella* species reported to cause Legionnaires’ disease. Aside from the well-characterized *L. pneumophila*, and to a lesser extent, *L. longbeachae*, very little is known of the mechanisms employed by non-pneumophila *Legionella* species in causing disease in humans. The use of plasmid-borne GFP is advantageous over that of immunofluorescence, as the latter requires cell fixation as well as specific primary antibodies that may not be available for non-pneumophila *Legionella* species, whereas the former allows real-time monitoring of bacterial growth dynamics in vitro and in vivo. While antibiotic supplementation for plasmid maintenance and metabolic burden of GFP expression may be negative aspects of the use of GFP, these requirements do not seem to significantly affect the growth of *L. bozemanae* in vitro and in vivo, as observed in this study. Thus, the use of GFP as a fluorescent marker is beneficial to future studies on non-pneumophila *Legionella* species. 

## Figures and Tables

**Figure 1 microorganisms-09-00379-f001:**
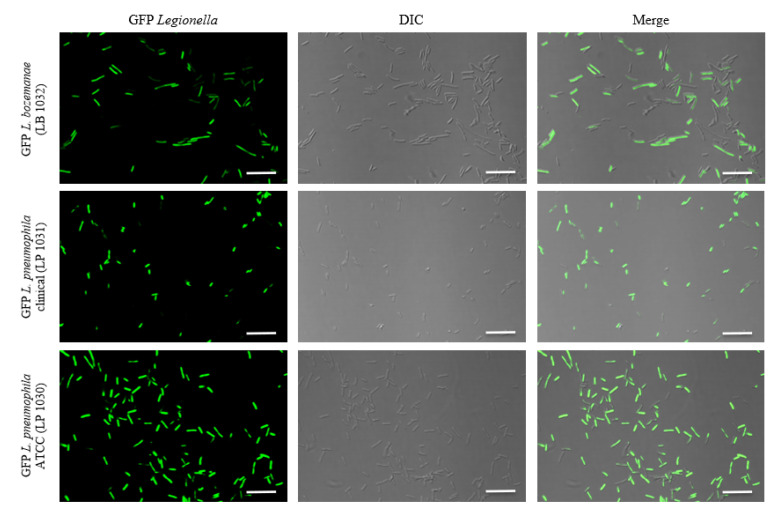
Differential interference contrast (DIC) and confocal microscopy of the GFP-expressing *L. pneumophila* and *L. bozemanae* strains imaged with a Zeiss LSM 700 confocal microscope. Images were generated using a 40× oil immersion objective. The panel on the far right represents the merged DIC and fluorescence channels. The scale bar represents 10 μm. ATCC, American Type Culture Collection; GFP, green fluorescent protein.

**Figure 2 microorganisms-09-00379-f002:**
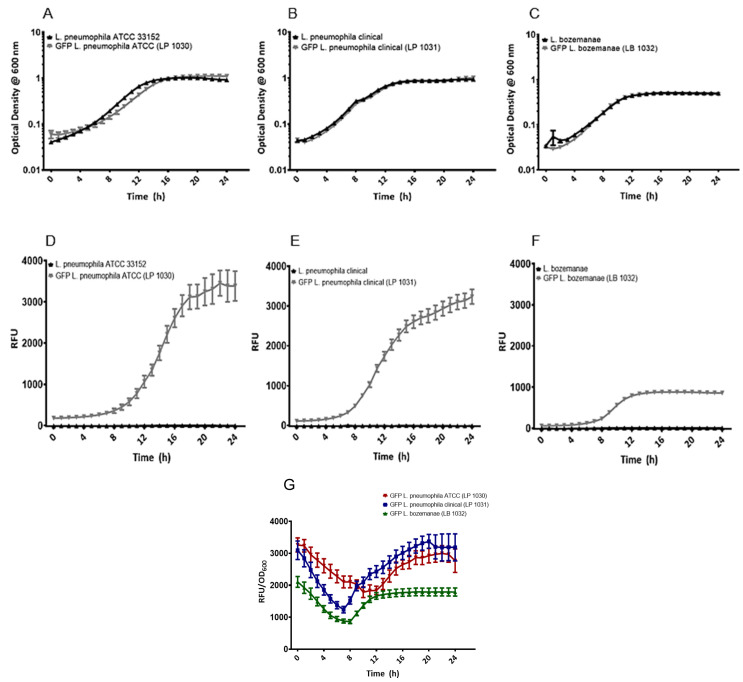
In vitro growth curves and relative fluorescence for the GFP-tagged *Legionella* and parental strains. All *Legionella* isolates were incubated in BYE broth with shaking in an automated microplate reader for 24 h where OD_600_ (**A**–**C**), RFU (**D**–**F**), and RFU/OD_600_ (**G**) were measured every hour. For panels **A**–**F**, the parental strains are displayed in black, while the GFP-expressing strains are shown in gray. For panel G, the GFP-expressing *L. pneumophila* ATCC and *L. pneumophila* clinical isolate are shown in red and blue, respectively, while the GFP-expressing *L. bozemanae* is in green. All experiments were performed in triplicate and data represent the mean ± standard errors; ATCC, American Type Culture Collection; BYE, Buffered Yeast Extract; GFP, green fluorescent protein; OD 600, optical density at 600 nm; RFU, relative fluorescent units.

**Figure 3 microorganisms-09-00379-f003:**
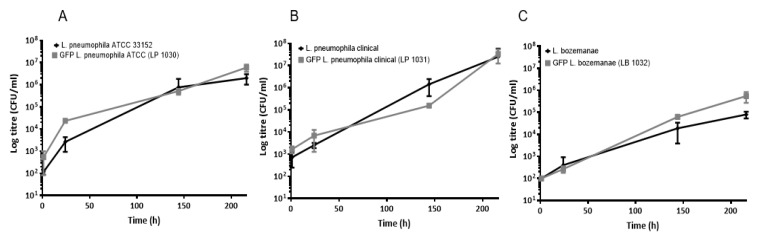
*In vivo* growth curves for the GFP-tagged *Legionella* and wild-type strains. Parental strains and GFP-tagged *L. pneumophila* ATCC 33152 (**A**), *L. pneumophila* clinical isolate (**B**) and *L. bozemanae* (**C**) strainswere grown in *Acanthamoeba castellanii* at 25 °C. Infections were performed at an MOI of 0.1 and *Legionella* intracellular growth was determined at 1, 48, 144, and 216 h post-infection. The parental strains are displayed in black, while the GFP strains are shown in gray. All experiments were performed in triplicate, and data represent the mean ± standard errors. ATCC, American Type Culture Collection; CFU, colony-forming units; GFP, green fluorescent protein; MOI, multiplicity of infection.

**Figure 4 microorganisms-09-00379-f004:**
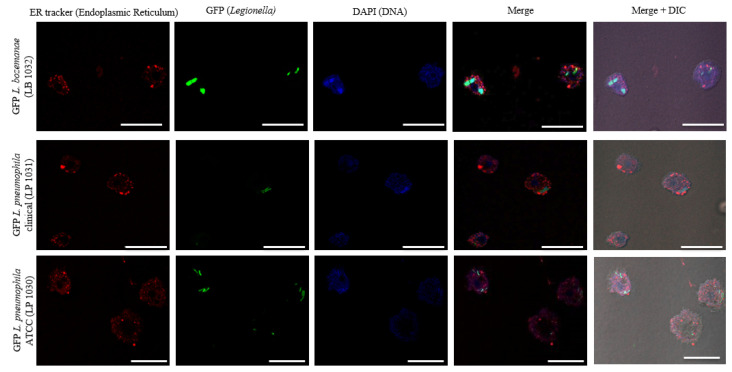
Confocal micrographs of *A. castellanii* infected with GFP-expressing *Legionella* strains. Image panels represent *A. castellanii* protozoa infected with GFP-expressing *L. bozemanae, L. pneumophila*, and *L. pneumophila* ATCC 33,152 strains 12 h post-infection visualized with a Zeiss LSM 700 confocal microscope. Individual fluorescence channels are displayed. The panel on the far right represents the merged fluorescence channels (green, GFP bacteria; red, endoplasmic reticulum; blue, DNA; and DIC). Scale bar = 10 μM. ATCC, American Type Culture Collection; DIC, Differential interference contrast; GFP, Green fluorescent protein.

**Table 1 microorganisms-09-00379-t001:** Organisms and plasmids used in this study.

Organisms	Genotype	Reference/Source
*Legionella pneumophila* ATCC 33,152 Philadelphia-1	Parental	ATCC
*Legionella pneumophila* 950 clinical isolate	Parental	Health Sciences Centre (Winnipeg, Canada)
*Legionella bozemanae* 948 clinical isolate	Parental	Health Sciences Centre (Winnipeg, Canada)
LP 1030	*L. pneumophila* ATCC 33,152 pGFP-Kan; Kan^r^	This study
LP 1031	*L. pneumophila* 950 pGFP-Kan; Kan^r^	This study
LB 1032	*L. bozemanae* 948 pGFP-Kan; Kan^r^	This study
*E. coli* DH5α	F’ *endA1 hsdR17(r_k_- m_k_-) supE44 thi-1 recA1 gyrA (Nal^r^) relA1 Δ(lacZYA-argF)U169 deoR*(Φ80lacΔ(lacZ)M15)	Laboratory stock (New England Biolabs)
*Acanthamoeba castellanii* (ATCC 30234)	Douglas	ATCC
**Plasmids**		
p34S-Kan	Cloning vector with Kan^r^ cassette	Laboratory stock Dennis et al., 1998 [33]
pKB127	245 bp P_magA_ region cloned into BamHI and XbaI sites of pBH6119; Amp^r^, Thy^+^	Morash et al., 2009 [26]
pGFP-Kan	pBH6119 P*magA*; Kan^r^	This study

Note: ATCC, American Type Culture Collection; Bp, base pair; GFP, green fluorescent protein; Kan^r^, kanamycin resistance; Thy^+^, thymidine auxotroph.

**Table 2 microorganisms-09-00379-t002:** List of primers and expected size of polymerase chain reaction products used in this study.

Primer	Sequence (5′ to 3′)	Annealing Temperature (°C)	Amplicon Size (bp)
pF KanR EcoRI	CGCATAgaattcCCACGTTGTGTCTCAAAATCTCTG	60	1120
pR KanR SacII	CGCATAccgcggGGTTGATGAGAGCTTTGTTGTAG	60	
pF KanR Conf Int	GTTGCATTCGATTCCTGTTTG	60	179
pR KanR Conf Int	GTGAGAATGGCAAAAGCTTATG	60	
pF KanR Conf Ext	GTGCCCATTAACATCACCATC	60	1200 ^a^
pR KanR Conf Rev	GTTATTTCTCCGGATTTAATTCG	60	726 ^b^

Note: ^a^ If used with pR Conf Int; ^b^ If used with pF Conf Int.; Underlining in a primer sequence indicates the restriction endonuclease cut site. bp, base pairs.

## Data Availability

Data is contained within the article or Appendix A.

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
