# Peer review of "Development of a Fluorescent Tool for Studying *Legionella bozemanae* Intracellular Infection"

_microorganisms, 2021, doi:10.3390/microorganisms9020379_

Round 1
Reviewer 1 Report
The aim of this publication is to detail the intracellular lifestyle of Legionella bozemanii within Acanthamoeba castellanii. To do this, the authors propose the use of a GFP plasmid in order to locate, post-incubation, Legionella bacteria, in particular L. bozemanii. The article demonstrates that transfection is relatively supported regardless of Legionella ssp.
The references mentioned are not always relevant. In some cases, the information mentioned is missing or incorrect:
- Line 36: [4-6] there is no information for Europe or Australia.
- Line 36-37: “In the United States (US), since 2000, Legionella incidence has risen almost 5-fold, with similar increases having been reported in Canada and Europe [7–9]”
However, the reference [7] indicates, in Europe:
- 5,835 reported cases in 2013
- 6,978 in 2014
- 6,972 in 2015 (+ 118%)
- 7,094 in 2016 (+ 102%)
- 9,238 in 2017 (+ 130%)
Likewise, line 39: “18,000 patient hospitalizations”; Collier et al. [10] indicate this value as the high value of an extrapolation, based on data from Ohio, carried out by Marston et al. in 1997.
Also, Collier et al. [10] specify “Legionnaires' disease, otitis externa, and NTM infection, diseases entirely or primarily transmitted by water, were estimated to occur each year and cost $430 million in costs for Medicare and Medicaid patients”. The amount indicated is therefore overestimated since Legionella are not the major agent of otitis externa and NTB infection.
Overall, this paper highlights many references without real justifications: [4-6; 12; 19-21; 23-25; 27-30; 44; 53]
References [1] and [6] are duplicates.
Reference [2] is a site marketing water management programs with a history of Legionnaires' disease cases referring to "suspected" or "possible" cases.
Figure 2G does not have a nomenclature associated with the graph.
There is no evaluation of the level of expression of the plasmid pGFP-Kan. From the images (Figure 1), the expression level is approximately 35% expression in L. bozemanii 1032 against 70 to 80% for L. pneumophilia 1030 and 1031
In conclusion, the authors present the article as, Line 18, “a comprehensive study detailing the intracellular lifestyle of an L. bozemanae” yet the information in the results and discussion sections is redundant.
The use of the GFP plasmid to locate a microorganism is a standard technique that is easy to perform.
There is no information on the interaction between L. bozemanii and A. castellanii, the infectious process, morphological, metabolic or other changes related to the intracellular lifestyle.
The title is clearly not in accordance with the data. This paper did not detail the intracellular lifestyle of Legionella bozemanii within Acanthamoeba castellanii. It should be presented as the development of a tool.
Reviewer 2 Report
The present manuscript is clearly written and the study methods and results appear methodologically sound and appropriate. The manuscript details the construction of a GFP plasmid and the successful transformation of a Legionella bozemanae clinical isolate (as well as strains of the more commonly studied L. pneumophila species) with this plasmid to generate GFP positive strains. The authors confirmed that GFP strains suffer no detectable metabolic burden in regards to in vitro and in vivo (in an amoeba host) growth rate, and observed the intracellular localization of all strains within amoeba. This work provides a pathway towards investigating additional non-pneumophila Legionella isolates through the ability to label and track them within host systems which will be useful in understanding the role of these isolates in clinical disease. The study is sufficient but could be more comprehensive with the addition of further quantitative (rather than qualitative) comparisons of the intracellular infection process or host impacts between the L. bozemanae and L. pneumophila strains. This could be done by quantifying number of intracellular bacterial cells at different time points post infection after exposure and some analysis of host fitness. If this is beyond the scope of the present study, I suggest the authors slightly modify some of their statements to reflect a smaller breadth of detail provided. For instance, the abstract states that the work is a “comprehensive study detailing intracellular lifestyle…” but it is probably more appropriate to simply say that it is a “study demonstrating the intracellular lifestyle…”
Reviewer 3 Report
A timely and useful contribution beyond establish methods for L. pneumophila to study a less well understood pathogen associated with LD. To generalize from the results presented, however, I have some concern in the temperatures used for incubations, noting that the optimum for the amoeba host used may not be 25 C, nor is it described what different growth rates may be for the two species, let alone strains of Legionella used (why use 37 C for a non-human evolved environmental pathogen)? So some further description and discussion of the effects of growth temperature are pertinent, given the single temperature comparisons used that infer faster and higher densities of intracellular growth of L. pneumophila compared to L. bozemanae (in the environment). While the +/- GFP-expressing stains used appear to not differ in growth, again, could perhaps be clearer at their optimal temperature(s) to illustrate no significant difference?
Minor comments:
L41, update with Collier, et al. Estimate of Burden and Direct Healthcare Cost of Infectious Waterborne Disease in the United States. Emerg Infect Dis 2021, 27, (1), 140-149.
L152, singular for media, so 'medium'
L153, only a broth once bacteria growing in it, so PYG 'liquid medium'
L202, replace 'Legionella can' with "L. penumophila can'; also the paragraph at L202-210 seem out of place for results, either discussion or perhaps better in the methods section?
Round 2
Reviewer 1 Report
Review MDPI 02/02/2021:
The aim of this revised version is now clearly established as the development of a fluorescent tool for studying Legionella bozemanae intracellular infection. To do this, the authors propose the use of a new GFP plasmid with an antibiotic gene to select transformants. The article demonstrates that transfection is relatively supported regardless of Legionella ssp.
The changes made including the modification of the title provide a better understanding of the work presented in the article.
Abstract L.21-22: Please change lifestyle by presence
Line 35: “In recent years, cases of Legionnaires’ disease have increased at an alarming rate […]”This passage should be qualified by the increase in the use of rapid and sensitive tests (molecular biology, etc.) like reference [6].
- Reference [6]: “An increasing population of older persons contributed to the increase in reported legionellosis cases. Other factors that might have contributed include an increasing population of persons at high risk for infection; improved diagnosis and reporting, possibly stimulated by the 2005 CSTE endorsement of more timely and sensitive legionellosis surveillance; and increased use of urine Legionella antigen testing. However, because increases in urine antigen testing began in the 1980s, its use is unlikely to account for the entire increase in legionellosis cases since 2000”
Line 37: “5-fold” however the reference indicates a lower incidence in the USA.
- Reference [6]: “For this report, cases reported to NNDSS during 2000--2009 from the 50 states and the District of Columbia (DC) were assessed, […] from 1,110 in 2000 to 3,522 in 2009 […].”
- Reference [5]: report from a single Canadian city indicating 6 cases in summer 2017. It is necessary to justify the evolution of the incidence of Legionellosis in Canada (national scale).
Line 51 to 55: one sentence, too long. Proposal:
The Legionella genus is comprised of more than 50 different species, with half of them able to cause disease in humans [12]. However, most available information about Legionella spp. pathogenesis and interaction with it host stems from studies focused on Legionella pneumophila serogroup (SG) 1, the type strain [13].
Line 58 to 61: one sentence, too long again. Proposal:
Indeed, studies from North America and Europe, regions which tend to favor the use of the UAT for diagnosing Legionella spp. infections, report L. pneumophila SG1 as the primary cause of LD (>90% of cases) [15]. This may be more reflective of the diagnostic methodologies rather than a true representation of species diversity.
Line 86: What about Legionella bozemanii for the presence of other plasmids? references?
Line 177-178: mechanical disruption (i.e. by scraping the monolayer and vortexing the amoeba): usually mechanical disruption of amoeba uses a gauge needle. The lysis efficacy should have been monitored by visual inspection (on Malassez chamber for example) because I guess that just vortexing might not be efficient. However, amoeba are not able to survive on BCYE plate and this should have led to bacteria release, but the number of bacteria may have been underestimated when several Legionella were included in the same amoeba cell (only one CFU for several Legionella).
Line 219 and 231: Figure 1 is present twice in the pdf version.
Line 236: DIC- Differential interference contrast should be suppressed at the end of the legend as it is already explained in the figure title.
Lines 264 and 304: data represents the mean ± standard errors of the means; According to the methods, it is simply the standard error (n=3), SEM would need a greeter number of replicates to extrapolate to the population.
Lines 330 to 332: “Transformation of the L. bozemanae strain, along with control L. pneumophila strains, with the modified pGFP-Kan plasmid by electroporation was achieved with relative ease.” Should be suppressed as it is already stated in the results (line 225) and as it is not really a discussion point.
Reference [18]: self-citation, work from 2007 to 2014 with 47 HIV patients associated with pneumonia. Among them 17 positive cases of Legionella infection were detected. Interest of the reference?
Reference [36]: self-citation, BYE growth instructions themselves reported from Feeley et al., 1979.
Reference [38]: self-citation, post-exponential expression already present in the reference [26].
Figure 2 – Conversion from word to pdf shows some failures, repeated letters, "G" not indicated.
Idem for figure 4: two figures seem superposed in the pdf version.
Author Response
Response to Reviewer 1- Round 2 Comments
Point 1: Abstract L.21-22: Please change lifestyle by presence
Response 1: We have made the change as you recommended.
Point 2: Line 35: “In recent years, cases of Legionnaires’ disease have increased at an alarming rate […]”This passage should be qualified by the increase in the use of rapid and sensitive tests (molecular biology, etc.) like reference [6]. Reference [6]: “An increasing population of older persons contributed to the increase in reported legionellosis cases. Other factors that might have contributed include an increasing population of persons at high risk for infection; improved diagnosis and reporting, possibly stimulated by the 2005 CSTE endorsement of more timely and sensitive legionellosis surveillance; and increased use of urine Legionella antigen testing. However, because increases in urine antigen testing began in the 1980s, its use is unlikely to account for the entire increase in legionellosis cases since 2000”
Response 2: We appreciate your feedback and have changed the section to the following:
“In recent years, cases of Legionnaires’ disease have increased at an alarming rate, with outbreaks having been reported across North America, Europe, and Australia [1–3]. Although the exact reason for this increase is unknown, it is likely multifactorial, representing increased disease awareness and diagnostic testing, in addition to an increase in the susceptibility of the global population, as well as global warming, and aging infrastructure, both of which favor Legionella growth in the environment.” (line 34, page 1)
Point 3: Line 37: “5-fold” however the reference indicates a lower incidence in the USA.
- Reference [6]: “For this report, cases reported to NNDSS during 2000--2009 from the 50 states and the District of Columbia (DC) were assessed, […] from 1,110 in 2000 to 3,522 in 2009 […].”
- Reference [5]: report from a single Canadian city indicating 6 cases in summer 2017. It is necessary to justify the evolution of the incidence of Legionellosis in Canada (national scale).
Response 3: We appreciate your comment. We have modified the section to the following:
“In Canada, Legionella incidence has risen 10-fold since 2000, with 635 cases having been reported to the Public Health Agency of Canada in 2018 [4]. Incidence has also increased 9-fold in the United States (US) and 2-fold in Europe [5,6].”(line 39, page 1)
We modified the US reference to one from the CDC website where they state that Legionella incidence has “grown by nearly nine times since 2000”. To better show that Legionella is increasing on a national scale in Canada, we changed our reference 5 to one form the Public Health Agency of Canada.
Public Health Agency of Canada. Legionellosis reported cases from 1986 to 2018 in Canada Available online: https://diseases.canada.ca/notifiable/charts?c=yl (accessed on Feb 2, 2021).
Point 4: Line 51 to 55: one sentence, too long. Proposal: The Legionella genus is comprised of more than 50 different species, with half of them able to cause disease in humans [12]. However, most available information about Legionella spp. pathogenesis and interaction with it host stems from studies focused on Legionella pneumophila serogroup (SG) 1, the type strain [13].
Response 4: We thank you for the proposed change and have taken your advice:
“The Legionella genus is comprised of more than 50 different species, with half of them able to cause disease in humans [5]. However, most available information about Legionella spp. pathogenesis and interaction with its host stems from studies focused on Legionella pneumophila serogroup (SG) 1, the type strain [6].” (line 63, page 2)
Point 5: Line 58 to 61: one sentence, too long again. Proposal:
Indeed, studies from North America and Europe, regions which tend to favor the use of the UAT for diagnosing Legionella spp. infections, report L. pneumophila SG1 as the primary cause of LD (>90% of cases) [15]. This may be more reflective of the diagnostic methodologies rather than a true representation of species diversity.
Response 5: We appreciate your suggestion and have changed the manuscript accordingly.
Point 6: Line 86: What about Legionella bozemanii for the presence of other plasmids? references?
Response 6: Publications regarding the presence of plasmid in L. bozemanae appears to be very few. There is a report of a strain of L. bozemanae carrying a large-sized plasmid (Aye et al.) so it is possible for this species to carry a plasmid. In this study, while it is not absolutely certain that L. bozemanae does not carry a plasmid, it is quite possible that it does not since it is able to harbor a RSF1010-based plasmid which is incompatible with another IncQ-derived plasmid. But, IncQ can co-exist with other plasmid groups such as ColE1. Nevertheless, the purpose of transforming L. bozemanae with GFP plasmid is to explore the possibility of using GFP expression as a tool to investigate non-pneumophila Legionella species. Future directions do include refinement of this approach by optimizing GFP expression through further genetic modifications of the plasmid as well as conducting whole genome sequencing to check for presence of any naturally-occurring plasmid.
Aye T, Wachsmuth K, Feeley JC, Gibson RJ, Johnson SR (1981) Plasmid profiles of Legionella species. Curr Microbiol 6:389-394.
Point 7: Line 177-178: mechanical disruption (i.e. by scraping the monolayer and vortexing the amoeba): usually mechanical disruption of amoeba uses a gauge needle. The lysis efficacy should have been monitored by visual inspection (on Malassez chamber for example) because I guess that just vortexing might not be efficient. However, amoeba are not able to survive on BCYE plate and this should have led to bacteria release, but the number of bacteria may have been underestimated when several Legionella were included in the same amoeba cell (only one CFU for several Legionella).
Response 7: We understand the reviewer’s concern regarding the chosen method of mechanical disruption. However, Dietersdorfer and coworkers evaluated the impact of methods used to release intracellular L. pneumophila from Acanthamoeba. In their article, they compared 8 methods, including mechanical and chemical disruption treatments, and they concluded that “…mechanical release treatments, namely three passages through a 21G or 27G needle or centrifugation at 10,000 × g for 10 min, were the most effective treatments, showing the highest recovery rates without risk of interference by chemical residuals.”
To clarify our methods, we expanded the statement to include more details:
“At various time points (1, 48, 144, 216 h post-infection), amoeba were lysed using mechanical disruption (i.e. by scraping the monolayer, vortexing the sample for 30 seconds followed by centrifugation for 8 min at 14,800 rpm) [37]…” (line 197, page 5).
We also referred ot the manuscript by Dietersdorfer et al.
- Dietersdorfer, E.; Cervero-Aragó, S.; Sommer, R.; Kirschner, A.K.; Walochnik, J. Optimized methods for Legionella pneumophila release from its Acanthamoeba hosts. BMC Microbiol. 2016, 16, 1–10, doi:10.1186/s12866-016-0691-x.
Point 8: Line 219 and 231: Figure 1 is present twice in the pdf version.
Response 8: Apologies, we are not sure why that happened in the PDF. We have gone through the new marked up version of the manuscript and have made sure that each Figure appears only once in the manuscript (right after the text that mentions them for the first time.
Point 9: Line 236: DIC- Differential interference contrast should be suppressed at the end of the legend as it is already explained in the figure title.
Response 9: Thank you for your attention to detail. We made the change.
Point 10: Lines 264 and 304: data represents the mean ± standard errors of the means; According to the methods, it is simply the standard error (n=3), SEM would need a greeter number of replicates to extrapolate to the population.
Response 10: Thank you for your comment. We have changed the lines as you suggested.
Point 11: Lines 330 to 332: “Transformation of the L. bozemanae strain, along with control L. pneumophila strains, with the modified pGFP-Kan plasmid by electroporation was achieved with relative ease.” Should be suppressed as it is already stated in the results (line 225) and as it is not really a discussion point.
Response 11: We appreciate your comment and have removed the statement. The line now reads, “Transformation did not appear to affect the growth kinetic profiles of the GFP-expressing Legionella strains compared to their parental counterparts.” (line 343, page 9)
Point 12: Reference [18]: self-citation, work from 2007 to 2014 with 47 HIV patients associated with pneumonia. Among them 17 positive cases of Legionella infection were detected. Interest of the reference?
Response 12: The interest of adding this reference was to show that in a study of individuals with HIV that presented with pneumonia, those that were infected with Legionella all had Legionella non-pneumophila SG1 infections.
Point 13: Reference [36]: self-citation, BYE growth instructions themselves reported from Feeley et al., 1979.
Response 13: We removed reference 36 and replaced it with the Feeley et al. 1979 manuscript.
Point 14: Reference [38]: self-citation, post-exponential expression already present in the reference [26].
Response 14: We removed reference 38 from the results section when referring to post-exponential expression as you suggested.
Point 15: Figure 2 – Conversion from word to pdf shows some failures, repeated letters, "G" not indicated.
Response 15: I’m not sure why the PDF appeared this way but I believe that the issues have been fixed in the new version of our manuscript.
Point 16: Idem for figure 4: two figures seem superposed in the pdf version.
Response 16: Again, we are not sure why that happened in the PDF. We have gone through the new marked up version of the manuscript and the issue appears to have been resolved.
